# Dephosphorylation of YB-1 is Required for Nuclear Localisation During G_2_ Phase of the Cell Cycle

**DOI:** 10.3390/cancers12020315

**Published:** 2020-01-29

**Authors:** Sunali Mehta, Cushla McKinney, Michael Algie, Chandra S. Verma, Srinivasaraghavan Kannan, Rhodri Harfoot, Tara K. Bartolec, Puja Bhatia, Alistair J. Fisher, Maree L. Gould, Kim Parker, Anthony J. Cesare, Heather E. Cunliffe, Scott B. Cohen, Torsten Kleffmann, Antony W. Braithwaite, Adele G. Woolley

**Affiliations:** 1Department of Pathology, University of Otago, P.O. Box 56, Dunedin 9016, New Zealand; sunali.mehta@otago.ac.nz (S.M.); cushla.mckinney@otago.ac.nz (C.M.); rhodriharfoot@gmail.com (R.H.); bhatia.puja17@gmail.com (P.B.); alistair.j.fisher@hotmail.com (A.J.F.); maree.gould@otago.ac.nz (M.L.G.); kim.parker@otago.ac.nz (K.P.); heather.cunliffe@otago.ac.nz (H.E.C.); antony.braithwaite@otago.ac.nz (A.W.B.); 2Maurice Wilkins Centre for Biodiscovery, University of Otago, Dunedin 9016, New Zealand; 3Centre for Protein Research, Department of Biochemistry, University of Otago, P.O. Box 56, Dunedin 9016, New Zealand; torsten.kleffmann@otago.ac.nz; 4Bioinformatics Institute (A*STAR), 30 Biopolis Street, 07-01 Matrix, Singapore 138671, Singapore; chandra@bii.a-star.edu.sg (C.S.V.); raghavk@bii.a-star.edu.sg (S.K.); 5School of Biological Sciences, Nanyang Technological University, 60 Nanyang Drive, Singapore 637551, Singapore; 6Department of Biological Sciences, National University of Singapore, 16 Science Drive 4, Singapore 117558, Singapore; 7Children’s Medical Research Institute, University of Sydney, NSW 2145, Australia; tara.bartolec@unsw.edu.au (T.K.B.); tcesare@cmri.org.au (A.J.C.); scohen@cmri.org.au (S.B.C.); 8Malaghan Institute of Medical Research, Victoria University, Wellington 6242, New Zealand

**Keywords:** YB-1, cell cycle, nuclear translocation, atomistic modelling, phosphorylation

## Abstract

Elevated levels of nuclear Y-box binding protein 1 (YB-1) are linked to poor prognosis in cancer. It has been proposed that entry into the nucleus requires specific proteasomal cleavage. However, evidence for cleavage is contradictory and high YB-1 levels are prognostic regardless of cellular location. Here, using confocal microscopy and mass spectrometry, we find no evidence of specific proteolytic cleavage. Doxorubicin treatment, and the resultant G_2_ arrest, leads to a significant increase in the number of cells where YB-1 is not found in the cytoplasm, suggesting that its cellular localisation is variable during the cell cycle. Live cell imaging reveals that the location of YB-1 is linked to progression through the cell cycle. Primarily perinuclear during G_1_ and S phases, YB-1 enters the nucleus as cells transition through late G_2_/M and exits at the completion of mitosis. Atomistic modelling and molecular dynamics simulations show that dephosphorylation of YB-1 at serine residues 102, 165 and 176 increases the accessibility of the nuclear localisation signal (NLS). We propose that this conformational change facilitates nuclear entry during late G_2_/M. Thus, the phosphorylation status of YB-1 determines its cellular location.

## 1. Introduction

Y-box-binding protein 1 (YB-1; encoded by *YBX1*), is a multifunctional protein involved in numerous biological processes including cell cycle progression, DNA repair, and drug resistance. There have been a number of reports linking YB-1 cleavage and subsequent nuclear translocation to poor cancer prognosis, reviewed in [1,2,3,4,5,6,7,8,9]. When in the nucleus, YB-1 acts as a transcription factor where it transactivates the growth promoting genes *c-ERBB2* [10] and *EGFR* [11] and also downregulates the death-promoting genes *TP53* [12] and *FAS* [13]. Nuclear translocation of YB-1 is reported to occur in a cell cycle dependent fashion [14,15] and in response to a range of stressors including DNA damaging agents [16,17,18]. As tumour cells are thought to be under constant stress due to the accumulation of mutations, the significance of nuclear YB-1 in cancer has been the focus of ongoing investigations. Nuclear YB-1 has been shown to be a negative prognostic marker in patients with a range of cancers including synovial sarcoma [19], breast [3], prostate [2] and non-small cell lung cancers [1]. However, other studies have found that it is the overall level of YB-1 protein (and mRNA), rather than its nuclear location, which is associated with high grade cancers [6,20,21,22].

Reports that increased nuclear YB-1 is linked to both tumour progression and drug resistance stimulated investigations into the molecular mechanism underpinning YB-1 transcriptional activation. A model of proteasome-mediated cleavage by the 20S proteasome through sequence-specific endoproteolytic cleavage was proposed [7,8]. Cleavage would allow the N-terminal region of YB-1 to be free of the dominant cytoplasmic retention signal (CRS; aa 247–267) [23], thus enabling the nuclear localisation signal (NLS; aa 186–205 [24]) to direct the cleaved N-terminal product to the nucleus (Appendix A). It was suggested that this proteolytic activation is associated with genotoxic stress, and that cleaved nuclear YB-1 is a distinct species with transcription factor activity compared to the full-length cytoplasmic YB-1 [7]. Subsequent domain mapping revealed the presence of three additional NLS at aa 149–156, 185–194 and 276–292 [9], with part of the latter located within the CRS (aa 264–290) previously proposed by Bader et al. [24]. Van Roeyen et al. also reported the presence of a C-terminal fragment in the nucleus following proteolytic cleavage [9], rather than the N-terminus, as previously reported [7]. We have sequenced nuclear YB-1 using mass spectrometry and found no evidence of cleavage at the aa 219/220 site [25]. Due to these inconsistencies within the literature we decided to further investigate whether we could detect any evidence of specific proteolytic cleavage. 

In this paper we used YB-1 plasmids with tags at each end of the protein and carried out immunofluorescent (IF) labelling after transfection of several cancer cell lines, either untreated or treated with doxorubicin (DOX), or paclitaxel (PTX). We also used confocal and live cell imaging and in some cases mass spectrometry of purified YB-1 protein. Our results provide no compelling evidence of specific cleavage at the site originally proposed in the 20S model [7,8]. We do however confirm that YB-1 migrates to the nucleus but we make the novel observation that this occurs during late G_2_/M coinciding with the onset of nuclear membrane disruption. Finally, we provide mechanistic evidence using 3D structural modelling, that the phosphorylation status of YB-1 alters the accessibility of both the cytoplasmic retention signal (CRS) and the nuclear localisation signal (NLS) and confirm this experimentally by showing that when these serine residues are mutated, YB-1 remains in the nucleus. We propose that dynamic changes in the phosphorylation status of specific residues of YB-1 and the resultant conformational fluctuation in the accessibility of both the NRS and the CRS, regulates the cellular location of YB-1.

## 2. Results

### 2.1. Full Length YB-1 is Present in Both Nuclear and Cytoplasmic Compartments

To determine whether YB-1 is full length or cleaved upon nuclear translocation we transfected three cancer cell lines (A549, H1299 and Saos-2) with a plasmid carrying both N- and C-terminal labels (*^HA^*YB-1*^FLAG^*, Appendix A) and quantitated the location of each tag using IF labelling of tag-specific antibodies. Confocal images of representative cells from each cell line are shown in Figure 1A. Both anti-HA (red) and anti-FLAG (green) antibodies detecting the N and C terminus of *^HA^*YB-1*^FLAG^* respectively, were found in the cytoplasm and the nucleus of all three cell lines and these two labels are co-localised (Figure 1A, lower panels). We quantitated the intensity of *^HA^*YB-1 and YB-1*^FLAG^* in at least 100 cells from each cell line. The nucleus and cytoplasm were defined using masks created by the imaging software (Figure 1B). Results from this analysis show that the average ratio of intensity per area of both *^HA^*YB-1 and YB-1*^FLAG^* is 1.0671 (range 0.8376–1.2966) for the three cell lines in either the nucleus or cytoplasm (Figure 1C), confirming that these two labels co-localise. 

This was shown further by performing western blot analyses on whole cell lysates as well as cytoplasmic and nuclear fractions. Full length YB-1 protein (49kDa) is present in both cellular fractions (Appendix A). There is no obvious evidence of enrichment for a smaller species (~32kDa) consistent with 20S proteasomal cleavage. 

In further confirmation of this conclusion, we performed LC-MS/MS analysis of endogenous YB-1 immunopurified from the cytoplasm and nuclei of A549 and MDA-MB-231 cells. Results show (Appendix A) that the peptides assigned to YB-1 from two independent runs cover the length of YB-1. Importantly, tryptic peptides (RPQYSNPPVQGEVMEGADNQGAGEQRPVR) spanning the proposed 20S proteasomal cleavage site (Appendix A) were identified in immunopurified YB-1 from cytoplasm (3/4 samples) and nucleus (3/4 samples; Appendix A). Mass spectrometry indicates that full length YB-1 is in the nucleus of A549 and MDA-MB231 cells. 

### 2.2. Specific Proteosomal Cleavage of YB-1 does not Occur in Response to Treatment with Doxorubicin or Paclitaxel

We next asked whether treatment with drugs causing genotoxic stress promotes YB-1 cleavage. We used either the topoisomerase inhibitor doxorubicin (DOX) or the microtubule inhibitor paclitaxel (PTX). We chose these two different treatments due to their differential effects on the cell cycle. DOX causes replicative stress leading to G_2_ phase arrest [26] and PTX causes cells to arrest in late G_2_/M and eventually the cells undergo death by mitotic catastrophe [27,28]. The dosages used for each of these drugs was chosen based on their IC_50_ values (Appendix A). Cells were treated with either drug 24 hours after transfection with *^HA^*YB-1*^FLAG^* and IF analysis was carried out. Again, both anti-HA (red) and anti-FLAG (green) antibodies detecting the N and C terminals of *^HA^*YB-1*^FLAG^* were found in the cytoplasm and the nucleus of all three cell lines (Figure 2A). Quantitation of >100 cells was performed as above and the results show that the average ratio of intensity per area of both *^HA^*YB-1 and YB-1*^FLAG^* is 1.13 (range 0.9017–1.3561) for the three cell lines treated with either DOX or PTX (Figure 2B). These data show that both labels are present in equal intensities, providing evidence that YB-1 is full-length. In order to ensure that this effect is not due to differences in antibody specificity, we repeated these experiments with the opposite secondary antibodies (i.e., Alexa Fluor 568 (red) anti-rabbit (HA) /Alexa Fluor 488 (green) anti-mouse (FLAG) compared to Alexa Fluor 488 (green) anti-rabbit (HA)/Alexa Fluor 568 (red) anti-mouse (FLAG). These data are shown in Appendix A; quantification of signal intensities in Appendix A. The average ratios of the 2 fluorochromes is 1.02 (range: 0.54–1.31) shown in Appendix A, essentially confirming the data in Figure 2B, implying that equal amounts of each tag are present in both compartments and arguing against proteolysis. There is one exception and that is with DOX treated nuclear YB-1 in the A549 cells where the ratio is ~0.5. suggesting relative loss of the C-terminus; but this was not consistent across experiments.

Cellular fractionation and immunoblotting confirm the presence of full length YB-1 in both nuclear and cytoplasmic compartments (Appendix A). The blots also reveal the presence of smaller YB-1 fragments. In both the A549 and H1299 cells they are primarily associated with the whole-cell (WCL) lysates. In the merged western blots for these cells, the fragments appear as green, suggesting that the C-terminal end of the protein (with the red tag) is no longer present. In the Saos-2 cells, additional fragments are evident in the nuclear lysates. Again, in the merged image these bands are green, suggesting the absence of the C-terminus. There is also a red band in the WCL fraction at ~20 kDa that is present following treatment with either PTX or DOX, suggesting the absence of the N-terminus (carrying the green tag). The size of these smaller bands is not consistent with that of a fragment proposed in the cleavage model [7]. To try and understand whether these fragments were influenced by the location of the tags, we carried out the same nuclear fractionation experiment with *^FLAG^*YB-1*^HA^* construct. These data are shown in Appendix A. Again, we find the presence of additional fragments in the WCL and nuclear fragments, which are more numerous in the H1299 cells. Again, none of these fragments is consistent with proteolytic cleavage at the 219/220 site. 

Of note, when we quantitated the cellular location of *^HA^*YB-1*^FLAG^* in response to DOX treatment, we surprisingly found many cells where YB-1 could not be detected in the cytoplasm (acytoplasmic; Figure 2C, D). As DOX leads to arrest in G_2_, this led us to consider the possibility that, in addition to the response to DNA-damaging agents, the nuclear location of YB-1 may be linked to cell cycle phase; and specifically, to the G_2_/M phases. 

### 2.3. Nuclear Localisation of YB-1 Occurs during Late G_2_ Phase of the Cell Cycle

To investigate this possibility, A549 cells were transfected with the *^HA^*YB-1*^FLAG^* plasmid and subsequently stained with antibodies to both HA, FLAG and the nuclear envelope marker Lamin B1 to determine YB-1 location. As shown in Figure 3A, the cellular location of YB-1, as demonstrated by IF analysis of the *^HA^*YB-1*^FLAG^*-transfected cell varies. Due to the fact that cells round up in early mitosis some of this variation appears to be linked to the cell cycle. To investigate this further, we carried out live cell imaging to identify cell cycle phases at single cell resolution using A549 FUCCI cells carrying both mCherry (red)-cdt1 and mVenus (green)-geminin tags [29]. By linking these two proteins to fluorescent labels, cell cycle progression can be visualised in real time [30]. These cells were transfected with a YB-1 construct carrying the blue fluorescent protein 2 (YB-1*^EBFP2^*) marker. The data (Figure 3B) show that during G_1_, YB-1 *^EBFP2^* is cytoplasmic (as shown by the red (cdt1) nucleus) but it begins to enter the nucleus in the G_2_ phase (as shown by the green (geminin) nucleus). During M phase, YB-1 *^EBFP2^* (blue) appears centrally in the cell (Figure 3B, arrowed). During the earliest stage of mitosis (prophase) the paired copies of DNA condense and the nuclear envelope begins to disintegrate, enabling the DNA pairs to align with the central axis of the cell. Towards the end of mitosis, as the cell moves into telophase, the two daughter cells move into G_1_ and YB-1 is again present in the cytoplasmic compartment (Figure 3B). 

To confirm that the observed movement of YB-1 from the cytoplasm to the nucleus during G_2_/M is not an artefact of exogenously expressed YB-1, we repeated this experiment in A549 cells that were synchronised using a double thymidine block and samples taken at 0, 1, 2, 3, 4 and 5 hours followed by 15 min time intervals up to 8 hours after release of the second block. These data are shown in Figure 4. The upper row of this figure is a projection of a z-stack of adjacent 0.2 µm optical slices acquired by confocal microscopy. The rows below represent top, middle and bottom slices of each z-projection. As previously reported, YB-1 can be found in both the nucleus and/or cytoplasm, and is often found in the perinuclear region of cells [4,20,22,31]. 

Interestingly, when YB-1 is perinuclear, when visualised using conventional light microscopy it often appears to be nuclear. Comparison between the z-projection of a cell in G_1_ (top left-hand panel) and the slices shown below provides insight into why YB-1 may appear to be nuclear when viewed two-dimensionally. The green punctate staining that appears nuclear on the projected stack in the first column is on the outside of the nuclear envelope. YB-1 is found in the cytoplasm during G_1_ (Figure 4; column 1) and increasingly localises to the perinuclear region during the G_2_ phase (Figure 4; column 2). We used this localisation, clearly visible by microscopy, as a marker of G_2_. As the cells progress into the mitotic phase of the cell cycle, YB-1 becomes localised to the nucleus. This is evident (Figure 4; column 3) in prophase which is the first phase of mitosis and continues until anaphase (Figure 4; column 6). Prophase is characterised by disruption of the nuclear envelope (as evidenced by lamin B1 staining (shown in red)). YB-1 remains nuclear until the nuclear envelope is re-established at the end of anaphase and begins to return to the cytoplasmic compartment during telophase (the last stage of mitosis; Figure 4; column 7).

### 2.4. Nuclear Localisation of YB-1 is Dependent on Phosphorylation Status 

Given that our data suggest that YB-1 nuclear translocation varies with cell cycle phase, we wished to understand how this process is controlled. Previously, phosphorylation of serine 102 [32,33,34,35], as well as phosphorylation of both serines 165 and 176 [36,37] have been shown to play a role in oncogenic activation. Given that two of these residues are close to the nuclear localisation signal (NLS; see Appendix A) we hypothesised that the phosphorylation status of these residues would alter the tertiary structure of YB-1 and hence the accessibility of the nuclear localisation signal (NLS). Increased accessibility of the NLS and/or reduced accessibility of the CRS would theoretically result in an increased amount of YB-1 present in the nucleus. To test this hypothesis, we performed atomistic modelling of YB-1 to measure the solvent accessibility surface area (SASA) in Å^2^ of the nuclear localisation signal (NLS), using the I-TASSER pipeline followed by molecular dynamics (MD) simulations. Of the five potential models that were generated by the I-TASSER server (Appendix A), model 3 was selected as the predicted cold shock domain of this model was close to the known NMR structure of YB-1 [38].

As shown in the graph at the far right of Figure 5A, the accessible surface area of the NLS in YB-1 where all three serine residues are phosphorylated (YB-1^phos^; black line) is reduced compared to unphosphorylated (YB-1^unphos^; red line). Thus, phosphorylation reduces access to the NLS, increasing the likelihood of YB-1 remaining in the cytoplasm. Modelling of individual serine mutations (S102A, S165A and S176A; red, brown and green lines) shows that the SASA values for the mutants are increased. Therefore, the accessibility of the NLS (shown in blue on each of the models) is dependent on the phosphorylation of these serine residues. Conversely, Figure 5B shows that phosphorylation of these serine residues (YB-1^phos^) leads to an increase in the accessibility of the cytoplasmic retention signal (CRS; shown in orange in the models). Again, perturbation of this phosphorylation in the mutants reduces the SASA value of the CRS. The predictions from this modelling were confirmed by IF labelling of HA-tagged YB-1 plasmid constructs containing alanine mutations at each of these three serine residues (S102, S165 and S176) transfected into either A549/FUCCI or HCT116 cells. The subcellular location of HA-tagged YB-1 was carried out by confocal analysis of Z-stacks using the colour-profiling tool in Fiji2. Representative confocal images of HA-YB-1 and each of the three HA-tagged YB-1 serine-to-alanine mutants are shown in Figure 5C, where nuclear localisation of HA-YB-1 (green) is clearly evident in the mutants. Quantification of the IF analysis in > 100 cells transfected the mutants (Figure 5C, graphs) reveals the number of cells with nuclear YB-1 is significantly increased (*p* < 0.0001) compared to wild type (wt)YB-1. The inability of these residues to be phosphorylated reduces the accessibility of the CRS and YB-1 remains nuclear. To determine whether this effect was seen in mutants where the acidic charge of the mutation was maintained, we carried out live cell imaging of cells where we had either mutated S176 alone or together with S165 to an aspartic acid (D) residue. Following transfection with the YB-1*^eBFP^*construct, live-imaging showed that YB-1 remains in the cytoplasm in contrast to the alanine mutants, in support of our hypothesis. Of note, these mutations led to cell death as they begin to enter mitosis (Appendix A).

Our results suggest a model in which dephosphorylation of YB-1 increases NLS exposure and leads to nuclear localisation, whereas YB-1 phosphorylation increases CRS exposure resulting in increased retention in the cytoplasm. To confirm that dephosphorylation of endogenous YB-1 does occur during the cell cycle, we carried out western blotting using either an antibody to wtYB-1 [25] or to pYB-1^S102^ on synchronised A549 cells. The timepoints that were chosen are based on the cell cycle progression of the A549 FUCCI cells following release from a double thymidine block, as shown in Appendix A. These data are shown in Figure 6; quantification of signal intensities contained in Appendix A. 

They confirm that despite levels of wtYB-1 remaining constant throughout the cell cycle (Figure 6A), phosphorylation levels of YB-1 as detected by the rabbit polyclonal against phosphorylated S102 (pS102) are increased during early S phase (0–5.5 h), followed by a decrease in the late S and G2 phases (6–7 h), and then an increase in M phase (9 h) (Figure 6B,C for quantitation relative to β actin). 

## 3. Discussion

Proteolytic cleavage has been reported to occur prior to nuclear translocation [7,8,9]. Elevated levels of YB-1 have been linked to oncogenic activity and poor prognosis in a range of cancers [1,2,3,4,5,6,20,22]. Previous analysis carried out by us does not support specific proteolytic cleavage at the aa 219/220 site [25]. The evidence regarding the details of proteolytic cleavage are contradictory with both N- and C-terminal fragments being reported as translocating to the nucleus by different groups [7,9]. To address these discrepancies, we use IF confocal analysis of three cancer cell lines transfected with a plasmid construct carrying epitopes to both an N-terminal HA and a C-terminal FLAG (*^HA^*YB-1*^FLAG^*) to determine whether this would provide evidence for cleavage. We find equal proportions of both epitopes suggesting that only full length YB-1 is present in the nucleus and the cytoplasm of cells and that this ratio does not change in response to treatment with either DOX or PTX. These results are confirmed by LC-MS/MS analyses of endogenous protein, where a tryptic peptide spanning the proposed site at aa 219/220 targeted by the 20S proteasome [5,7,9] was detected. Western Blot analysis of subcellular fractions did reveal the presence of smaller fragments. However, the size of these fragments is not consistent with the predicted proteasomal cleavage product, which would run at ~32 kDa [7,9]. 

Specific cleavage was originally reported to occur in response to treatment with bovine α-thrombin in endothelial cells in vitro [23]. Subsequently, cleavage via the 20S proteasome was reported to take place in YB-1 purified from rabbit reticulocyte lysate, in murine fibroblasts [7] and in rat mesangial cells [9], in response to a range of treatments including DOX. However, neither cleavage nor nuclear localisation occurred following treatment with taxol [7], a result supported by our data. Of note, the concentrations of DOX that we used were based on the IC_50_ values (see Appendix A) which is ~two orders of magnitudes less (0.005 µg/mL) than the 0.6 µg/mL previously used [7,9]. In our hands the percentage of surviving cells at the higher dose was drastically reduced, as shown in Appendix A. The use of this lower dose is in line with that used previously [26,39]. The differences between our results and those previously published [7,9,23] may be due to both the difference in the DOX concentration and to the response of the different cell lines used in these studies. It is also possible that the tags we used (HA and FLAG) could alter putative site-specific cleavage patterns. However, this seems unlikely as Sorokin et al. showed that cleavage of ectopically expressed HA-YB-1 by the 20S proteasome produced both 32 kDa and 22 kDa protein species [7] in untreated and DOX treated cells and the generation of these two species was blocked following treatment with the proteasome inhibitor MG132. Similarly, van Roeyen et al. demonstrated that proteasomal cleavage of GFP-tagged YB-1 takes place in response to DOX treatment [9]. Both of these reports illustrate that neither the N-terminal nor C-terminal tags interfere with either cleavage or nuclear import of YB-1.

Our finding that the cellular location of YB-1 is cell cycle dependent is in line with earlier reports [14,15,21,40,41,42]. YB-1 has been shown to transcriptionally activate mRNAs for the cyclins *CCNA1* and *CCNB1* [14] as well as *CDC6* [40]. This finding was further extended by bioinformatic analysis to reveal that YB-1 binds to the promotor regions of E2F target genes including *CDC6*, *CCND1* and *CDK1* [21] in human cancer cells, thus driving cell cycle progression. Most recently, Kotake et al. (2017) used a *Rb* family triple knockout (TKO) model to investigate the role of YB-1 in cell cycle phases other than G_1_. They showed that in this model silencing YB-1 led to a G_2_/M phase arrest and the repression of *CCND1* [15]. In the same paper, RIP-CHIP assay showed YB-1 to be associated with the mRNA of multiple cell-cycle genes including that of G_2_/M phase regulators [15]. These findings are in line with our finding that YB-1 moves into the nucleus during the G_2_/M transition, potentially to bind with mRNAs identified by Kotake et al [15]. In line with this, nuclear YB-1 has been shown to interact directly with the cyclin A2 promoter leading to a downregulation of cyclin A2 mRNA expression in zebrafish [42]. In addition to this transcriptional role, YB-1 may also play a post-translational regulatory role. These same authors demonstrate that presence of YB-1 in the nucleus is driven by a circadian clock mechanism and involves the SUMOylation of YB-1 [42]. YB-1 has been shown to be ‘nuclear’ in 5% of exponentially growing HeLa cells, 6–8 hours following release from lovastatin-induced G_1_ arrest [14]. As lovastatin activates p21^CIP1^ and p27^ink4a^ cyclin inhibitory proteins at or near the G_1_/S phase boundary [43,44], the timing would be consistent with our data that nuclear translocation of YB-1 occurs in late S phase or G_2_ and then into mitosis (M). This is further supported by a subsequent paper [41], where the authors found that tetracycline-induced YB-1 overexpression for a sustained period of time led to increased expression of cyclin E and subsequent slippage through the G_1_/S checkpoint [41]. In the same paper, IF analysis found that YB-1 localised to the centrosome forming a complex with pericentrin and α-tubulin which was vital in maintaining structural integrity and microtubule nucleation capacity of the organelle. These reports suggest that YB-1 plays a principal role in the organisation of the cytoskeleton during cell division. Our data support such a role. 

The importance of YB-1 phosphorylation at serine 102 has long been known since it was recognised as key in the transcriptional activation of oncogenic genes such as *EGFR* [34]. S102 has been shown to be a target of p90 Ribosomal S6 Kinase (RSK) [35,45]. Phosphorylation of either serine 165 or 176 of YB-1 has also been reported to activate NF-κB [37]. The atomistic modelling data presented here defines a novel mechanistic role for serine residues 102, 165 and 176. Our data show that the phosphorylation status of these residues is dynamic and alters the conformation of YB-1 providing a mechanism for biomolecular recognition as detailed in [46]. Such phosphorylation-induced conformational switching has also been reported in PAGE4, another intrinsically disordered protein [47]. 

We propose a model, as shown in Figure 7, where the serine residues of YB-1 are phosphorylated in G_1_ leading to an increase in the accessible energy surface area of the CRS, as suggested by the 3D structural modelling in Figure 5A. YB-1 remains cytoplasmic as the cells move from G_1_ through to G_2,_ when dephosphorylation occurs and YB-1 begins to move into the nucleus in late G_2_ just prior to the onset of nuclear envelope disruption in early M (prophase), concomitantly reducing the accessibility of the CRS and increasing the accessible energy surface of the NLS resulting in nuclear translocation. These residues are again phosphorylated in late M (telophase), to facilitate its export to the cytoplasm where it now (presumably) plays a role in translational regulation [48,49,50,51]. 

This model supports previously published data [7], where removal of the C-terminal region of YB-1 was shown to lead to nuclear translocation. We suggest that the loss of the C-terminal region may lead to a similar conformational change in YB-1 making the NLS more accessible. It is also in agreement with data demonstrating that nuclear translocation of YB-1 is dependent on an intact NLS [52].

## 4. Materials and Methods

### 4.1. Cell Culture

Cells obtained from American Type Culture Collection (ATCC, Manassas, VA, USA) were cultured in either Dulbecco's Modified Eagle Medium (DMEM)-F12 (A549, Saos-2, MDA-MB-231 and HCT116) or RPMI (H1299), Life Technologies (Thermo Fisher Scientific), Waltham, MA, USA or Sigma (Merck), St Louis, MO, USA supplemented with 10% fetal bovine serum (FBS), in a 37 °C humidified incubator under 5% CO_2_. The cell lines were validated for authenticity by CellBank Australia (http://www.cellbankaustralia.com/) using STR profiling and were regularly tested and found negative for mycoplasma contamination.

### 4.2. Generation of ^HA^YB-1 ^FLAG^ and ^FLAG^YB-1^HA^ Expression Constructs 

A haemaglutinin (HA) tag flanked by XbaI and BamHI sites was cloned into the N-terminal of *YBX1* cDNA of a pc3-DNA backbone using a pcDNA3.3-TOPO cloning kit (Thermo Fisher Scientific). Serine to either alanine or aspartic acid mutations at specific serine residues (S102, S165, S176) were generated in the pcDNA3.3 *HA-YBX1* plasmid by site-directed mutagenesis using mutagenesis primers (Appendix A) and were subsequently amplified with a Kapa high fidelity polymerase (Kappa HiFi PCR kit #KK2502, Roche Diagnostics, Rotkreuz, Switzerland). To introduce a FLAG tag to the 3’ end of *HA-YBX1*, two unique restriction sites, an EcoR1 site near the 3’ end of *YBX1* and an Xho1 site, were used for cloning in conjunction with the TOPO-TA kit (Thermo Fisher Scientific). The DNA removed by these nucleases was replaced with synthetic oligonucleotides. The oligonucleotides contained compatible ends to replace the 3’ end of *YBX1* with the original sequence while incorporating an in-frame FLAG tag before the termination codon. 

### 4.3. Generation of YB-1^EBFP2^ Expression Constructs 

A two-stage process was used to create HA-tagged YB-1*^EBFP2^* expression constructs. Firstly, an oligonucleotide encoding an HA tag flanked by XhoI and HindIII 5’ and SacII/BamHI 3’ was cloned into the XhoI/BglII sites in the MCS of pEBFP2-N1 (Addgene 54595) in-frame with EBFP2. The coding sequence of YB-1 and upstream Kozac sequence was amplified using TaKaRa Hi Fidelity Taq and cloned upstream of the HA tag XhoI/HindIII. The pHA-YB-1 was generated with pcDNA3.3-TOPO TA Cloning Kit (Thermo Fisher Scientific). The resulting plasmid produced a YB-1-HA-EBFP2 (YB-1*^EBFP2^*) fusion protein under the control of the CMV promotor. All plasmids were propagated through Stbl3 *E. coli* to prevent recombination and sequenced prior to experimental use.

### 4.4. Generation of A549 FUCCI Cell Line 

FUCCI was used to identify cell cycle phases at a single cell resolution. mVenus-hGeminin (1/110)/pCSII-EF and mCherry-hCdt1 (30/120)/pCSII-EF (a kind gift from Atsushi Miyawaki and Hiroyuki Miyoshi) were individually packaged into lentivectors using the 2nd generation packaging system, and the viral supernatants were used simultaneously to co-infect target cells. Three days post-transduction, cell cultures were sorted at the Westmead Institute for Medical Research flow cytometry core (Sydney, Australia) for mVenus fluorescence, allowed to expand for 5 to 7 days, and sorted again for mCherry fluorescence. Proper progression of red/green coloration during cell cycling was confirmed with live cell imaging as described below before use. 

### 4.5. Plasmid Transfection and Drug Treatment

Cells were transfected using Lipofectamine 3000 (LF3000, Invitrogen (Thermo Fisher Scientific, Waltham MA, USA), distributed by Biosciences, Dublin, Ireland)) according to the manufacturer’s instructions. For drug treatment, media were changed 24 hours post transfection, the cells were then treated with either 100 nM Paclitaxel or 10 nM Doxorubicin in DMSO for 24 hours. DMSO was used as a control for drug treatment. The medium containing the drug (or control) was then aspirated and the cells were thoroughly washed in PBS prior to collection for cell fractionation for either western blotting (6-well plates), fixation with 4% paraformaldehyde (24-well plates), or immunofluorescence (IF).

For live cell imaging A549 FUCCI cells (1 × 10^4^) were seeded in 2mL of DMEM supplemented with 10% FBS in a 12 well plate. After overnight incubation, cells were transfected using Lipofectamine 3000 (LF3000, Invitrogen) with 250 ng of shRNA targeting the 3’UTR (sh-YB-1) in combination with 250 ng YB-1*^EBFP2^*. Media was changed 24 hours post transfection following which the cells were monitored over 60 hours using the live cell time lapse imaging or were harvested at 48 h for western blot.

### 4.6. Live Imaging and Processing of Time-Lapse Data

For live cell imaging, cells were grown in 12-well glass bottom plates (MatTek, Ashland, MA, USA). Time lapse live cell imaging was performed on a Zeiss Cell Observer inverted wide field microscope, with 20 × 0.8 NA air objective, at 37 °C, 10% CO_2_ and atmospheric oxygen. Image capture commenced 24 hours post transfection, with images taken every six minutes for a duration of sixty hours using an Axiocam 506 monochromatic camera (Zeiss, Oberkochen, Germany) and Zen Blue software (v1.0, Zeiss). A Zeiss HXP 120C mercury short-arc lamp and compatible filter cubes were used to obtain fluorescent images and differential interference contrast (DIC) microscopy to capture brightfield images. To achieve optimal image resolution without excessive illumination, a binning factor was applied prior to imaging and the ambient conditions were maintained to minimize variations in optical resolution and illumination. The acquired videos were analyzed using the Zen Blue software (v1.0, Zeiss). For all videos, mitotic duration and outcomes were scored by eye and calculated from nuclear envelope breakdown until cytokinesis. FUCCI videos were scored by eye, for G_1_ (red) and S/G_2_/M (green) and EBFP2 (blue).

### 4.7. Cell Fractionation and Western Blotting

Cells were harvested for cell fractionation using a Cell Fractionation kit (Cell Signaling Technology, cat #9038, Danvers, MA, USA) according to the manufacturer’s instructions with the exception of substitution of 4× NuPage LDS for the 3× SDS loading buffer. Protein samples were then electrophoretically separated on Bolt™ 4–12% Bis-Tris Plus Gels (Invitrogen) using MES buffer (LiCor, Lincoln, NA, USA) and an equal amount of total protein was loaded into each lane. The Precision Plus Protein Dual Color Standard Ladder (Bio-Rad NZ, Auckland, NZ) was used. Proteins were transferred onto nitrocellulose membranes using iBlot2 gel transfer stacks (Invitrogen). Membranes were blocked with Odyssey^®^ Blocking Buffer (PBS) (LiCor) for 1 h and then incubated for 1 h with the primary antibodies diluted in Odyssey^®^ Blocking Buffer (PBS)/0.2% Tween20. For the HA/FLAG western experiments shown in Appendix A and S4 the primary antibodies used were rabbit anti-HA (H6908 Sigma-Aldrich 1:500 and mouse anti-FLAG (Sigma-Aldrich F1804 1:500, (Merck St Louis, MO, USA), with rabbit anti-H3 (Millipore, 1:5000, Burlington, MA, USA)) and mouse -tubulin (Cell Signaling Technology, 1:5000) as loading controls. For the westerns to detect changes in serine 102 phosphorylation (Appendix A), TBS blocking buffer was used together with goat anti-YB-1 [25] or rabbit anti-phospho-YB1^ser102^(Cell Signalling Technology) 1:1000 and mouse anti-β-actin, clone AC-15 (Sigma-Aldrich, 1:10000) as the loading control. In both sets of experiments, membranes were subsequently incubated with the secondary antibodies (IRDye^®^ 680RD Goat anti-Mouse IgG (LiCor) and IRDye^®^ 800CW Goat anti-Rabbit IgG (LiCor)) diluted in Odyssey^®^ Blocking Buffer (TBS or PBS)/0.2% Tween20. Membranes were imaged on the Odyssey^®^ CLx Imaging System (LiCor) according to the manufacturer’s instructions. Images were quantified using ImageStudioLite (LiCor) software.

### 4.8. Cell Synchronization and IF Labelling 

Cells were seeded onto coverslips in a 24 well plate and subsequently subjected to a double thymidine block with 2 mM thymidine (Sigma). Following 7–8 hours release from the second block, cells were then fixed in 4% PFA, permeabilised with 0.2% Triton X-100 (Sigma-Aldrich) and then blocked with 10% goat or donkey serum in PBS (depending on the secondary; Sigma-Aldrich) prior to incubation with primary antibodies in blocking solution. Primary antibodies used were rabbit anti-HA (Sigma-Aldrich H6908, 1:1000); mouse anti Flag M2 (Sigma-Aldrich F1804, 1:1000); rabbit anti-LaminB1 (Abcam 16048, 1:250) and sheep anti-YB-1 polyclonal to an N-terminal epitope (MSSEAETQQPPAC) of YB-1 generated in house (Cohen et al. 2010); 1:1000). Following primary incubation, cells were washed with PBS-Tween20 (0.1%) to remove any unbound primary antibody and then incubated with either Alexafluor-labelled secondary antibodies (goat anti-mouse 568/goat anti-rabbit 488 or goat anti-rabbit 568/goat anti-mouse 488, and (in the case of the sheep anti-YB-1) donkey anti-goat 568, all at 1:1000). In some instances, the DNA-binding dye Hoechst 33258 (Molecular Probes 1:1000) was used to visualise cell nuclei. Cells were then washed with PBS prior to mounting in Fluoromount-G (SouthernBiotech, Birmingham, AL, USA) on glass slides prior to visualization using a Nikon Ti-E inverted microscope with a 63× oil objective (N/A 1.4) and equipped with a DSRi2 CCD camera (Nikon Belmont, CA, USA). The acquired images were analysed using both NIS elements (Nikon) and Fiji2. In addition, a Lionheart FX Automated Live Cell Imager (BioTek, Winooski, VT, USA) equipped with a 20 × 0.45 NA air objective was used to collect data from multiple cells on these slides (at least 100 cells per image) using the montaging function of Gen5 4.0 from both 365nm, 488nm and 568nm channels. Images were then analysed for fluorescence intensity using the Dual masking capability of the same software.

### 4.9. Cell Proliferation Assay

Cells were treated with paclitaxel (1–1000 nM) or doxorubicin (0.01–30nM) for 48h, at which time the media was removed and the plates were frozen at −80°C. The plates were subsequently thawed and the DNA content measured using a SYBR Green I-based fluorimetric assay as described previously [53]. Briefly, 100 µL of lysis buffer (10mM Tris-HCl pH 8.0, 2.5mM EDTA and 1% (vol/vol) Triton X-100) containing 1:4000 SYBR Green I (Invitrogen) was added to the wells and the plates were incubated overnight at 4 °C. The plates were mixed and the fluorescence signal for each well was measured for 1 second at an excitation of 485nm and emission of 535nm using a BioTek microplate reader (BioTek). Growth curves were plotted as the fluorescence values at drug concentration.

### 4.10. Subcellular Fractionation of Cultured Cells for LC-MS/MS

Harvested cells (A549 and MDA-MB-231) were enriched into subcellular fractions as described previously [25]. Briefly, 1 × 10^4^ cells per µL were swollen in a hypotonic buffer (10 mM HEPES, pH 7.9, 1.5 mM MgCl_2_, 10 mM KCl, 0.5 mM DTT, 1 × Complete EDTA-free), and incubated for 5 min at 4 °C before being transferred to a Dounce homogeniser (885300-0002, 2 mL, tight pestle; 0.013–0.064 mm clearance, Kontes, Wayne, PA, USA). The Dounce was applied until 95% of the cell membranes were ruptured and lysate was spun at 220 *g* for 5 min at 4 °C. The supernatant was collected and stored as cytoplasmic extract. The pelleted nuclei were resuspended in 5 mL of chilled low sucrose solution (0.25 M sucrose, 10 mM MgCl_2_) which was gently layered over 5 mL of chilled high sucrose solution (0.88 M sucrose, 10 mM MgCl_2_). The nuclei were spun at 2800 *g* for 10 min at 4 °C to remove contaminating cytoplasmic proteins.

### 4.11. Immunopurification of YB-1 and Mass Spectrometry

YB-1 was purified from the nuclear and cytoplasmic fractions of two cancer cell lines; the adenocarcinoma cell line A549 and the breast cancer cell line MDA-MB23, as previously described [25]. Purified YB-1 was then separated using SDS-PAGE and stained with Coomassie (Appendix A). Excised bands were analysed by LC-MS/MS to identify the proteins that copurified with immunoprecipitated YB-1, again as previously described [25]. Peaklists and MS/MS data were generated using the Proteome Discoverer (version 1.4) software using default settings and searched against the Human Swiss-Prot amino acid sequence database using both the Mascot (http://www.matrixscience.com) and Sequest (Thermo Scientific) search engines. The search was set up for full tryptic peptides with a maximum of 3 missed cleavage sites. Carboxyamidomethyl cysteine, oxidized methionine, deamidation (N, Q), and phosphorylation (S, T, Y) were included as variable modifications. The precursor mass tolerance threshold was 10 ppm and the maximum fragment mass error 0.8 Da. Peptides were accepted as identified if their false discovery rate adjusted scores were above the score threshold at a false discovery rate≥ 1% as assessed by the Percolator algorithm [54]. Proteins were considered identified when they were assigned ≥ 2 peptides above the aforementioned score threshold. 

### 4.12. siRNA Transfection

Cells were reverse transfected with stealth modified 25 bp duplex siRNAs targeted to YB-1 (si-YB-1 [21]); or a scrambled control siRNA (si-Control [53] primer sequences detailed in Figure 6 legend) both from Invitrogen. The control siRNA has no known human mRNA targets [53]. Stealth siRNAs were transfected at a final concentration of 5 nM using Lipofectamine RNAiMax (Invitrogen). Both siRNAs and RNAiMax were diluted in medium without serum. After 10 minutes at room temperature, the diluted RNAiMax was added to the siRNAs, and the mixture was incubated for a further 15 minutes. The lipoplexes formed were added to cells. After overnight transfection, the culture medium was replaced with medium supplemented with 10% FBS following which the cells were harvested at 48 h for western blot.

### 4.13. Modelling the Structure of YB-1 Protein

In the absence of the crystal structures of full length YB-1 protein, the 3D structure was generated using standard protein modelling methods. The full-length sequence of YB-1 was used as query to retrieve structures of sequences with homology to YB-1 from the protein databank database using the program BLASTp [55], however no template structure was identified for the full length protein. Therefore, the automated I-TASSER pipeline [56,57], which uses multi-threading alignment and iterative template fragment assembly simulations to generate a 3D structural model of the YB-1 protein. 

### 4.14. MD Simulations

The structures of full length wild type (wt) YB-1, phosphorylated YB-1 (YB-1^phos^), YB-1^S102A^, YB-1^S165A^, and YB-1^S176A^ were modelled and subjected to molecular dynamics (MD) simulations using the *pemed.CUDA* module of the program Amber18 [58]. The all atom version of the Amber 14SB force field (ff14SB) [59] was used to model the proteins. Force field parameters for phosphorylated serine were taken as described elsewhere [60]. The phosphate groups were assigned an overall charge of −2e. The *Xleap* module of Amber18 was used to prepare the system for the MD simulations. All the simulation systems were neutralized with appropriate numbers of counterions. Each neutralized system was solvated in an octahedral box with TIP3P [61] water molecules, leaving at least 10 Å between the solute atoms and the borders of the box. All MD simulations were carried out in explicit solvent at 300K. During the simulations, the long-range electrostatic interactions were treated with the particle mesh Ewald [62] method using a real space cutoff distance of 9 Å. The Settle [63] algorithm was used to constrain bond vibrations involving hydrogen atoms, which allowed a time step of 2 fs during the simulations. Solvent molecules and counterions were initially relaxed using energy minimization with restraints on the protein and inhibitor atoms. This was followed by unrestrained energy minimization to remove any steric clashes. Subsequently the system was gradually heated from 0 to 300 K using MD simulations with positional restraints (force constant: 50 kcal mol^-1^ Å^-2^) on the protein over a period of 0.25 ns allowing water molecules and ions to move freely. During an additional 0.25 ns, the positional restraints were gradually reduced followed by a 2 ns unrestrained MD simulation to equilibrate all the atoms. For each system, three independent MD simulations (assigning different initial velocities) were carried out for 250 ns with conformations saved every 10 ps. Solvent accessible surface area (SASA) of the nuclear localization signal (NLS; residues 183–205) and cytoplasmic retention signal (CRS; residues 247–267) was calculated using the program NACCESS [64]. Simulation trajectories were visualized using VMD [65] and figures were generated using PyMOL [66]. 

## 5. Conclusions

Using quantitative confocal analysis of cells fluorescently labelled with a construct tagged at both termini (*^HA^*YB-1*^FLAG^*), we found equal ratios of both termini present in cytoplasmic and nuclear compartments in both drug-treated and untreated cells, suggesting that cleavage does not take place. In confirmation, we then used mass spectrometry to reveal the presence of a tryptic peptide spanning the proposed cleavage site. We also revealed, using live cell imaging, that the cellular location of YB-1 is linked to cell cycle phase. Full-length YB-1 is primarily cytoplasmic and begins to enter the nucleus as cells transition through the G_2_/M phase of the cell cycle, coinciding with the dissolution of the nuclear membrane. Towards the end of mitosis, with the reassembly of the nuclear membrane, YB-1 is again excluded to the cytoplasmic compartment. 

To provide insight into the mechanism underpinning the cellular location of YB-1 we then used atomistic modelling data and molecular dynamics simulations to show that the phosphorylation status of particular serine residues affects the conformation of YB-1 altering the accessibility of both the nuclear localisation signal (NLS) and the cytoplasmic retention signal (CRS). We confirmed the validity of this model by showing that YB-1 is retained in the nucleus in cells with a mutation of either serine S102A, S165A or S176A, by confocal z-stack analysis. We also showed, using live-cell imaging that mutation of S176 to S176D either independently or together with mutation of S165 to S165D lead to an inability of YB-1 to enter the nucleus at mitosis. These findings are consistent with the model. 

## Figures and Tables

**Figure 1 cancers-12-00315-f001:**
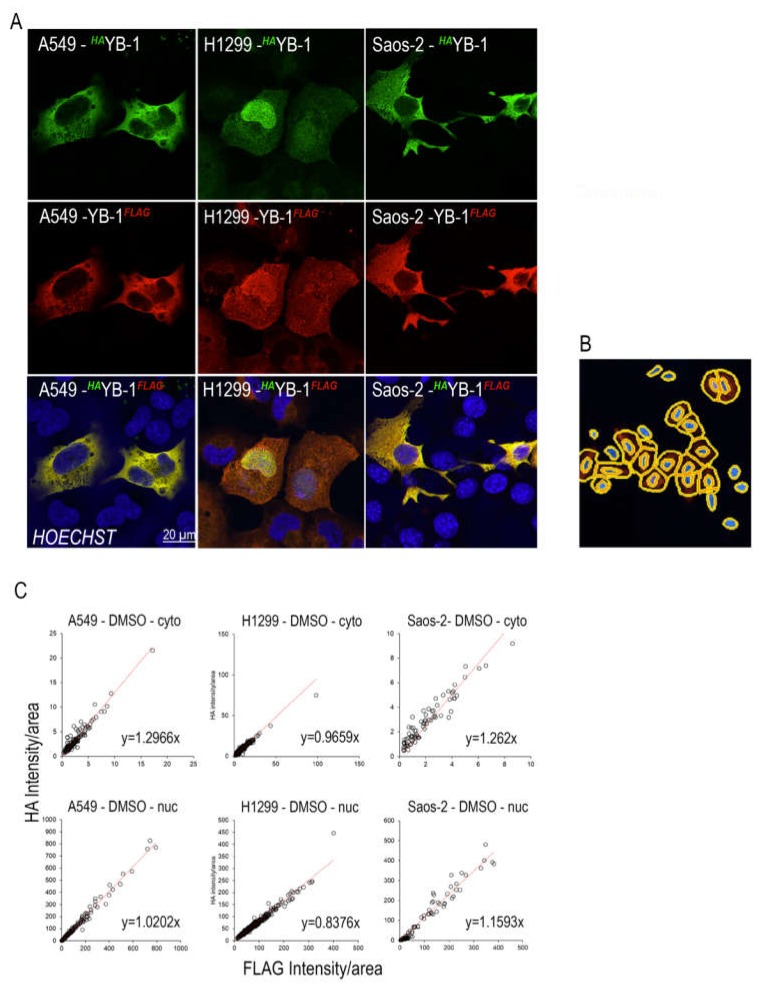
Full length YB-1 is present in the nuclear and cytoplasmic compartments. (**A**) Shows the nuclear and cytoplasmic location of *^HA^*YB-1*^FLAG^* in A549, H1299 and Saos-2 cells. Top row shows anti-HA which detects the N-terminus of *^HA^*YB-1*^FLAG^*, middle row shows anti-FLAG which detects the C-terminus of *^HA^*YB-1^FLAG^ and the bottom row shows the merged image. The nucleus is stained with Hoechst, anti-HA (green) and anti-FLAG (red). (**B**) shows examples of the nuclear and cytoplasmic mask used to determine cellular location of *^HA^*YB-1*^FLAG^*. (**C**) the fluorescence intensity/area for anti-HA and anti-FLAG using this mask for > 100 cells in all three cell lines. Each dot represents the intensity/area for anti-HA and anti-FLAG for each cell and the slope of the line is indicated.

**Figure 2 cancers-12-00315-f002:**
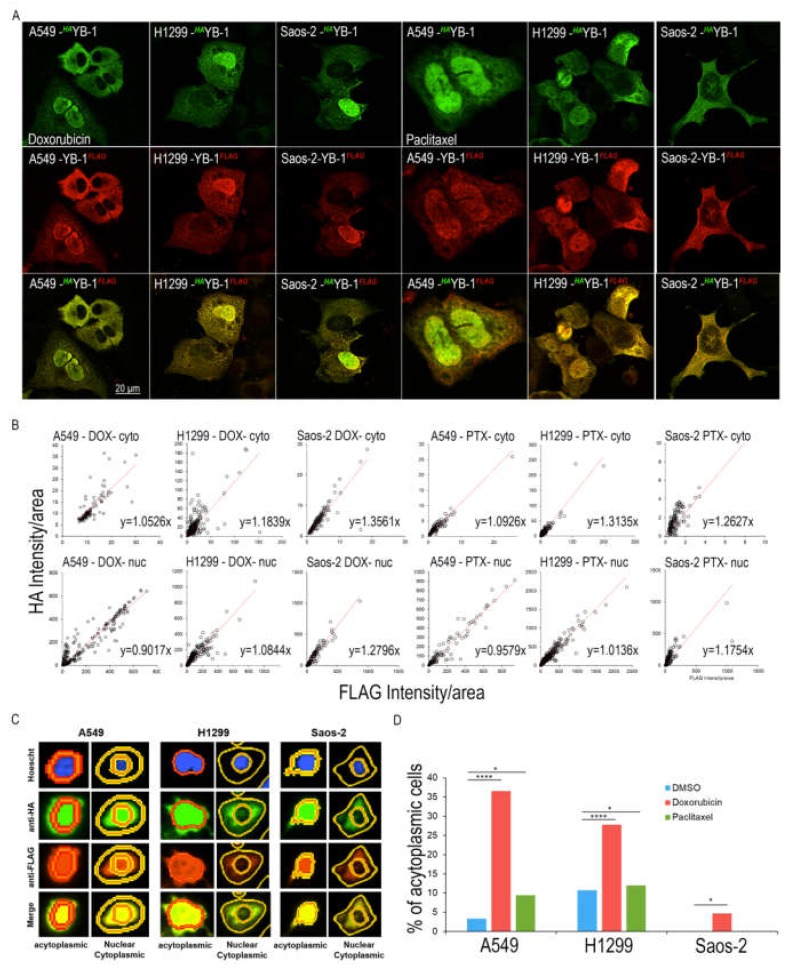
Full length YB-1 is present in both compartments following DNA damage. (**A**) shows the nuclear and cytoplasmic location of *^HA^*YB-1*^FLAG^*in A549, H1299 and Saos-2 cells treated with DOX or PTX. Top row shows anti-HA which detects the N-terminus of tagged *^HA^*YB-1*^FLAG^*, middle row shows anti-FLAG which detects the C-terminus of tagged *^HA^*YB-1*^FLAG^* and the bottom row shows the merged image. The nucleus is stained with Hoechst, anti-HA (green) and anti-FLAG (red). (**B**) the fluorescence intensity/area for anti-HA and anti-FLAG using masks to differentiate the nucleus and the cytoplasmic compartments for > 100 cells in all three cell lines. Each dot represents the intensity/area for anti-HA and anti-FLAG for each cell. The red dotted line represents the line of best fit. The slope of the line is indicated. (**C**) shows example of the masks with either the presence or absence of cytoplasmic staining of *^HA^*YB-1*^FLAG^* using anti-HA and anti-FLAG in A549, H1299 and Saos-2 cells treated with DOX. (**D**) Shows the percentage of acytoplasmic cells; i.e. those that did not show cytoplasmic labelling with either anti-HA or anti-FLAG, indicating that YB-1 was not present in the cytoplasm of these cells. Chi-square test was used to determine significance, * *p* < 0.05, **** *p* <0.0001.

**Figure 3 cancers-12-00315-f003:**
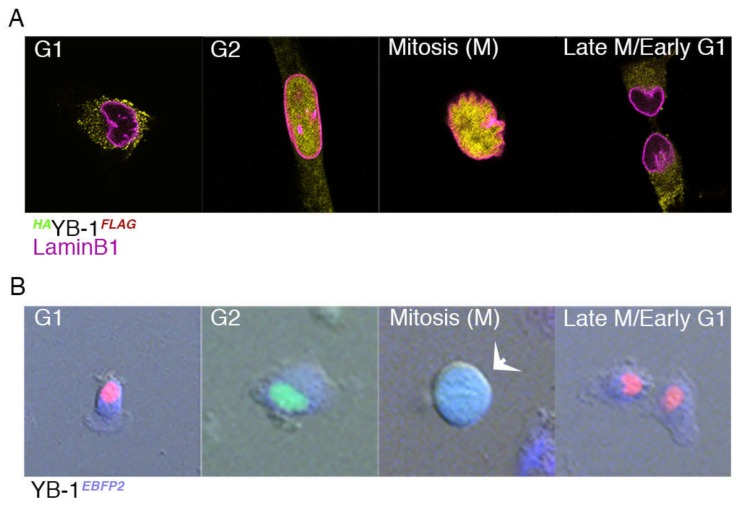
YB-1 translocates into the nucleus in the late G_2_ phase of the cell cycle. (**A**) Mid z-stack slices (0.2 µm) showing cellular location of *^HA^*YB-1*^FLAG^* (yellow) in A549 cells. LaminB1 (magenta) marks the nuclear envelope. (**B**) Illustrates the cellular location of YB-1*^eBFP2^* in different phases of the cell cycle in A549 FUCCI cells transfected with YB-1*^eBFP^*(blue); G_1_ marked by cdt1(red), S/G_2_ marked by geminin (green). The white arrow shows a mitotic cell where YB-1 (blue) fills the whole cell.

**Figure 4 cancers-12-00315-f004:**
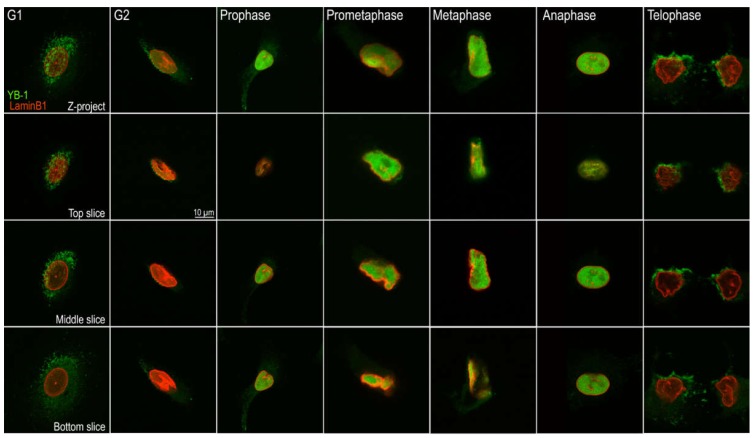
Cellular location of YB-1 is dependent on cell cycle phase. The top row are z-projections of 0.2 µm slices of synchronised A549 cells at sequential phases of the cell cycle. The second row shows the top slice, the third row shows the middle slide and the last row shows the bottom slices of each stack respectively. YB-1 (shown in green) is cytoplasmic in G_1_, becomes increasingly perinuclear during G_2_ and moves into the nucleus during prophase and remains there until telophase when it appears to be outside the nucleus. Lamin B1 is shown in red. Scale bar: 10 μm.

**Figure 5 cancers-12-00315-f005:**
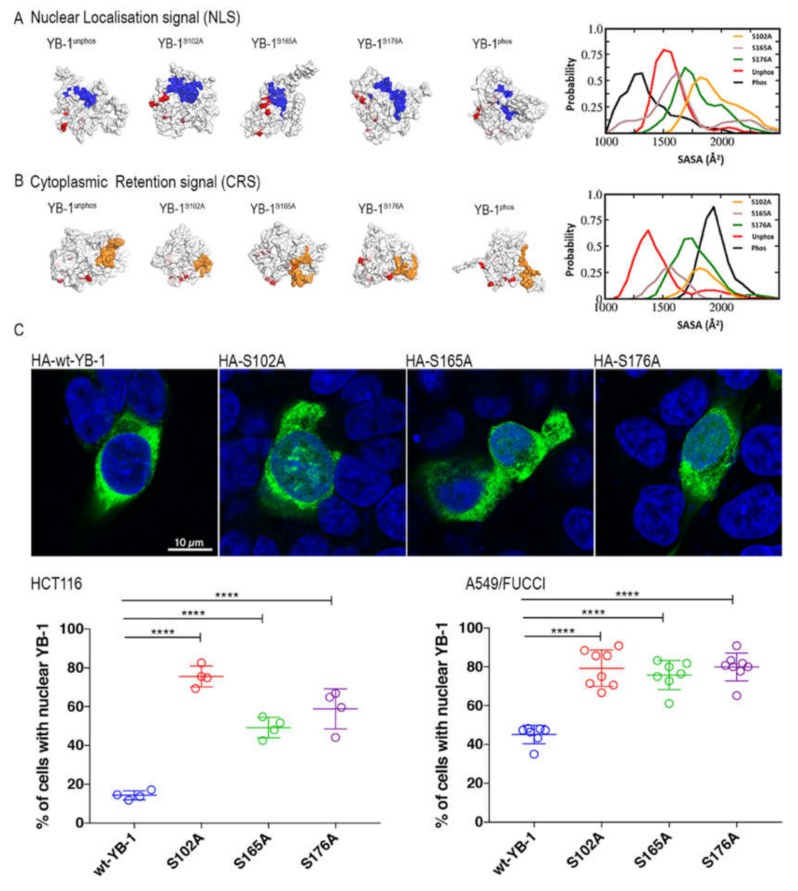
Phosphorylation of serine residues affects both NLS and CRS accessibility. (**A**) the solvent accessible surface area (SASA; blue) of the nuclear localisation signal (NLS; residues 183–205) in (from left to right) YB-1^unphos^, where the serine residues are not phosphorylated; YB-1 with serine 102 mutated to an alanine (YB-1^S102A^); YB-1^S165A^; YB-1^S176A^. A model of the NLS SASA when all serine residues are phosphorylated (YB-1^phos^) is shown on the right. The graph on the far right shows the SASA range for each of the models generated with NACCESS. (**B**) the solvent accessible surface area (SASA; orange) of the cytoplasmic retention signal (CRS; residues 247–267) in (from left to right) YB-1^unphos^, where the serine residues are not phosphorylated; YB-1 with serine 102 mutated to an alanine (YB-1^S102A^); YB-1^S165A^; YB-1^S176A^. A model of the CRS SASA when all serine residues are phosphorylated (YB-1^phos^) is shown on the right. The graph on the far right shows the SASA range for each of the models generated with NACCESS. **C,** shows (from left to right) confocal images of HCT116 cells with HA-tagged wild type (wt)YB-1 (green); HA-tagged YB-1^S102A^; YB-1^S165A^ and YB-1^S176A^. Cell nuclei are stained with Hoechst (blue). The graphs on the right-hand side show the quantification of the percentage of cells with nuclear YB-1 in both HCT119 and A549_FUCCI cells. Each circle represents one experiment where at least 100 cells were analysed. The increase in the percentage of cells with nuclear YB-1 between those transfected with wtYB-1 or with a plasmid containing serine to alanine mutations, in both cell lines is highly significant (*p* < 0.0001).

**Figure 6 cancers-12-00315-f006:**
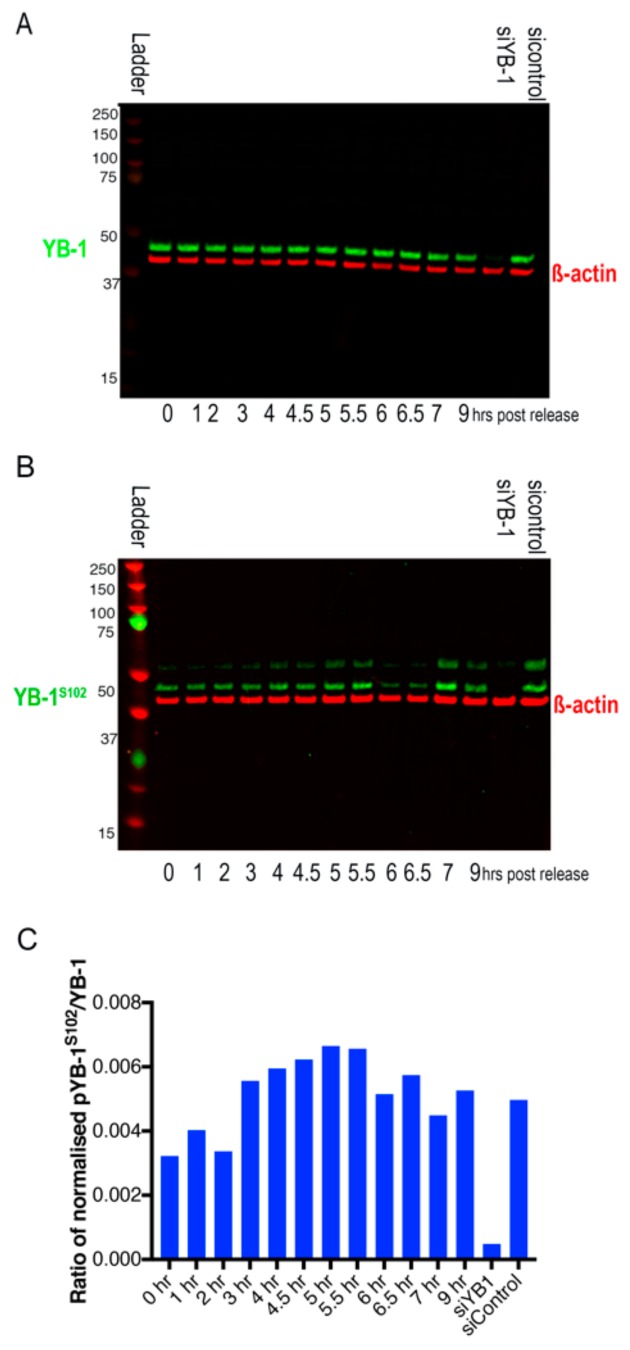
Western Blots from A549 cells following release from double thymidine block. Timepoints at which cells were harvested are shown below each of the blots. (**A**) YB-1, as detected by rabbit anti-YB-1 polyclonal antibody is shown in green. β-actin, as detected with a mouse anti-β-actin antibody is shown in red. (**B**) YB-1^S102^, as detected by rabbit anti-phospho-YB1^ser102^ (Cell Signaling Technology, Danvers, MA, USA) is shown in green. β-actin, as detected with a mouse anti-β-actin antibody is shown in red. The last two lanes show treatment with si-YB-1 5′-GGUCCUCCACGCAAUUACCAGCAAA-3′) or a scrambled si-Control 5’-CCACACGAGUCUUACCAAGUUGCUU-3 from Invitrogen. (**C**) Ratio of pYB-1^S102^/YB-1 in A549 cells post 2T block. Densitometries for these westerns is shown in Appendix A (Appendix A).

**Figure 7 cancers-12-00315-f007:**
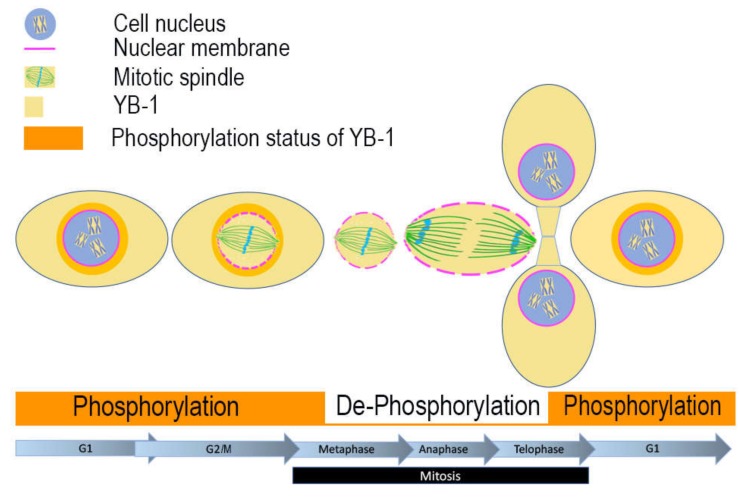
Proposed model of YB-1 location during cell cycle phases. YB-1 is shown in yellow. The phosphorylation status of YB-1 is depicted by the orange blocks below the cell graphic. In G_1_ phosphorylated YB-1 is cytoplasmic and becomes increasingly perinuclear as the cell moves from G_1_/S/G_2_. Just prior to the breakdown of the nuclear membrane in late G_2_ (represented by the broken magenta line), YB-1 is dephosphorylated, increasing the accessibility of the NLS enabling its movement into the nucleus where it remains during mitosis until the nuclear envelope is re-established in late telophase. Following phosphorylation and increased accessibility of the CRS, YB-1 moves to the cytoplasmic compartment. The mitotic spindle is depicted in green; chromosomes in light blue.

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
