# Peer review of "Dephosphorylation of YB-1 is Required for Nuclear Localisation During G2 Phase of the Cell Cycle"

_cancers, 2020, doi:10.3390/cancers12020315_

Round 1
Reviewer 1 Report
In this study, the authors present data indicating that:
(1) YB-1 nuclear translocation does not require proteolityc cleavage,(2) YB-1 subcellular localization is linked to cell cycle progression, (3) YB-1 nuclear localization depends on its phosphorylation status.
The first two points have been addressed by other authors, Jurchott K. JBC 2003 and Cohen SB Oncogene 2010; however, I agree with the authors that they are still controversial.
The use of a YB-1 plasmid with a tag to each end of the protein is an elegant approach to demonstrate that YB-1 translocates to the nucleus as full-length protein. The third point is totally new.
Although the data are interesting they raise me some points that it might be worth to address better.
The Doxorubicin treatment causes both DNA damage and G2 cell cycle arrest. Given that YB-1 is known to be recruited on DNA damage foci (Kim ER Cell cycle 2013; Fomina EE. Biochemistry 2015); how can the authors affirm that YB-1 nuclear accumulation upon doxorubicin treatment is only due to cell cycle arrest rather than to DNA damage?
The authors use two different treatments to have differential effects on the cell cycle: Dox (G2 phase arrest) or PTX (G1 arrest). According to recent manuscripts, similar to DOX, PTX induces cell death by halting cells in G2/M phase and both induce cell apoptosis (Andrea Ghelli Luserna di Rorà Journal of Hematology and Oncology 2019). Instead of using DOX or PTX, it should be possible to synchronize the cell population by serum deprivation and quantify cytoplasmic and nuclear YB-1. Alternatively, cell cycle-specific state of cells can be enriched by cell sorting.
DNA damage causes full-length YB-1 protein degradation (di Martino et al. Genes to Cells 2016). How the authors can exclude that DOX and PTX treatment do not affect the YB-1 protein level? Apparently, in Figure S2B, Saos2 panel, YB-1 fragmentation can be detected in total extracts only upon Dox and PTX treatment. Should it be possible to quantitate these blots using nuclear and cytoplasmic markers?
According to the arrest in the G1 phase claimed by the authors, By using PTX we should expect a decrease of acytoplasmic (nuclear?) cells compared to control after PTX treatment? The results presented in the plot of Figure 2D do not support this conclusion. Again, to address this question, I would suggest synchronizing cells by serum deprivation and quantify cytoplasmic YB-1.
Concerning point 3 (YB-1 nuclear localization depends on its phosphorylation status) I think it should be important to provide data that the stability and half-life of generated YB-1 mutant proteins (S102A, S165A, S176A) are not affected by mutations.
In conclusion, this is an interesting study and most of the data indicating the absence of 20S proteolytic cleavage of YB-1 and the relevance of the phosphorylation status for YB-1 for subcellular localization are relevant. However, the results on the subcellular localization of YB-1, using DOX or PTX, are confusing because of the pleiotropic effect of these agents on cell cycle and DNA damage and require additional experiments to warrant their conclusion. Finally, data from literature indicate that YB-1 nuclear accumulation also depends on DNA damage stimuli or circadian rhythm entrainment and that YB-1 polyubiquitination and degradation can occur in the nuclear compartment and be regulated. I would suggest to the authors mention these works in the introduction and discuss them in the context of the data presented.
Minor point
Figure 2A the label of line 1 panel six should be Saos-2 YB-1FLAG.
Reviewer 2 Report
This manuscript presents data on the mechanism which regulates the cytoplasmic-nuclear localization of the multifunctional protein YB-1 that is implicated in various biological process such as cell cycle progression, DNA repair and drug resistance. Its nuclear translocation has been associated with poor prognosis in cancer.
Mehta and colleagues present data on three different human cell lines showing the presence of the YB-1 full length form in both the cellular compartments independently of the presence or absence of genotoxic stressors. They conclude that the YB-1 nuclear translocation is not due to a previously postulated cleavage event, but due mainly to dephosphorylation events occurring during specific phases of cell cycle progression (G2-M).
The data are interesting in that they give clarification to the controversial issue of the regulation of YB-1 nuclear localization. The novelty of this paper is the involvement of dephosphorylation events in regulating YB-1 nuclear translocation. Although the manuscript is well written, and the experimental procedures are sound and elegant, this study needs some revision before it can be suitable for publication.
Major Revision:
The authors showed that the full-length Y-B1 construct is present in both cytoplasm and nucleus in non-treated as well PTX and DOX treated cell lines. The data seems convincing but the conclusion that the average ratio of intensity of the two signals (Flag and HA) is similar, it is based on the assumption that the two antibodies have the same relative affinity for the corresponding epitopes. I think the authors need to repeat the experiment also with independent antibodies for the two tags. In addition, the western blot presented in Figure S2 lacks quantification analysis for the total, as well for the cytoplasmic and the nuclear extracts.The authors state that small products present in the whole extract do not change in intensity after treatment with DOX or PTX and do not represent cleavage products but are only the products of degradation. This affirmation is not convincing. In figure S2 it is clear that when these low molecular bands are present in one cell line, they appear in every sample with the same molecular weight. This is not consistent with randomly cleaved degradation products. The intensity of these bands seems also increased following treatment (see H1299 with PTX or A549 with DOX or Saos-2 using the HA-tag). In addition, these extra bands are never present using the Flag-antibody except in the case of the Saos-2 cells were one band that is not recognized by the HA- antibody increases after treatment. Again, the quantification of these western blots and an analysis of the presence and intensity of these products are important to clarify this point.
The authors state that when they treat cells with DOX, they “found many cells where YB-1 could not be detected in the cytoplasm (acytoplasmic; Figure 2C-D)", but they do not comment or explain why also after PTX treatment there is a statistically significant increase of YB-1 acytoplasmic cells and not a reduction as expected from a drug that block cells in G1/S (Fig 2D).It will be very interesting to compare this data with control data documenting the number of cells effectively blocked in G2 or G1 after DOX and PTX treatment.
The authors show with an elegant experimental approach that the increase of nuclear YB-1 is associated with the G2 / M transition and during G1 phase YB-1 is mainly cytoplasmic. However, in their previous experiment with PTX they found less cytoplasmic YB-1 with respect to non- treated cells (Figure S2b). How they explain this data? This point needs to be discussed.
The authors need to expand the discussion that seems too simplistic and short. There are other papers showing the relationship between the cell cycle and the shift between nuclear and cytoplasmic YB-1 that have been ignored. For example, Basaki et al. (European Journal of Cancer, 2010), Kotake et al. (Anticancer Research, 2017) or Pagano et al. (Oncotarget, 2017). In the latter, it has been demonstrated that the circadian clock regulates the cell cycle progression via enrichment of the nuclear fraction of YB-1 during M-phase which inhibits cyclin B2 transcription.
Minor revisions:
In Figure S1B, the label of each line is confused with no alignment between the name of the cell lines and the correspondeing loading position. Please label properly. In Figure 2A the panels with the red green IFs referred to Saos-2 cells are inverted. Why in Figure 1A are two panels presented for A549 cells? If there is a reason please explain it in the text or in the figure legend.
Reviewer 3 Report
YB-1 is a multifunctional nuclear-cytoplasmic DNA- and RNA-binding protein that is involved in a number of cellular and physiological events, both normal and pathological, including cancers. In the cytoplasm, YB-1 participates in the regulation of mRNA translation, while in the nucleus, it regulates transcription of many genes and is involved in pre-mRNA splicing and DNA repair. It is considered to be a cancer marker displaying oncoprotein properties; its overexpression and/or nuclear localization is directly associated with cancer severity because it elevates the cell division rate, stimulates the multiple drug resistance, and increases metastasis. The nuclear translocation of YB-1 was previously shown to occur at the G1/S boundary of the cell cycle, under the influence of UV radiation and DNA-damaging xenobiotics, as well as during a certain phase of the circadian rhythm. It was proposed that the YB-1 transition from the cytoplasm to the nucleus may be regulated by two mechanisms: 20S proteasome cleavage of the YB-1 C-terminal sequence containing the cytoplasmic retention signal (CRS) and YB-1 phosphorylation at Ser102 with AKT- or RSK kinase. The precedence of either mechanism may depend on the type of inducer, the type of cells, and the cellular context.
In this study, the authors are focused on YB-1 nuclear translocation under the action of xenobiotics (anticancer drugs) and during the cell cycle. Their conclusions are, firstly, that in three lines of cancer cells, the doxorubicin (DOX)- or paclitaxel (PTX)-induced YB-1 translocation to the nucleus is unassociated with proteasomal cleavage of the protein, and secondly, that at the G1/S boundary, YB-1 does not enter the nucleus but becomes increasingly perinuclear. Its nuclear translocation occurs at a later stage of the cell cycle (G2/M boundary) and believed to be associated with dephosphorylation of S102, S165, and S176 because it is enhanced when these residues are replaced with alanine. Atomistic modeling and molecular dynamics simulations allowed the authors to suggest that dephosphorylation of S102, S165, and S176 increases the accessibility of the nuclear localization signal (NLS) through changes in the YB-1 conformation, and this entails the transition of YB-1 to the nucleus at the G2/M boundary. In contrast, phosphorylation of these amino acid residues increases the accessibility of CRS, thus promoting the export of YB-1 from the nucleus after mitosis.
Regretfully, many of the above results are incomplete due to the lack of proper controls or improper methods of investigation. When studying the mechanism of YB-1 transition to the nucleus under the action of doxorubicin (DOX) or paclitaxel (PTX), the authors use three approaches: (i) mass spectrometric analysis of proteolytic fragments of endogenous YB-1 (Fig. S1 (C)), (ii) determination of the ratio between two fluorescent labels to N- and C-terminal tags of the plasmid-synthesized form of YB-1 (Fig 1A, B), and (iii) electrophoretic analysis of nuclear and cytoplasmic YB-1 with tags (Fig. S2 (B)). In the third case, as well as in the second, YB-1 was detected using fluorescent antibodies to its tags: green to the N-terminal and red to the C-terminal. According to the electrophoretic analysis, the bulk of YB-1 displayed the electrophoretic mobility of a full-length protein and showed staining with both green and red antibodies. But in addition to the full-length YB-1, we can see higher mobility bands stained in green that show the presence of C-truncated YB-1 whose localization is (as expected) mostly nuclear. The authors believe that these bands cannot result from the action of the proteasome because of their molecular weights that had been calculated from their mobility relative to the markers. However, no direct comparison of the electrophoretic mobility of these bands with that of 20S proteasome-truncated YB-1 was made.
According to the authors, the second approach failed to reveal the truncated form of YB-1. Consequently, this integrated analysis of YB-1 based on the ratio between two types of tag-binding antibodies is not sensitive enough and allows no right conclusions. The same is true for the analysis of proteolytic fragments of YB-1 by mass spectrometry. Unfortunately, in the latter case, the authors presented no original data, but only the results of their analysis.
In summary, the authors' conclusion about the absence of C-terminal cleavage of YB-1 in the studied cells contradicts their most reliable experimental data. Here, we would recommend adding another control to test the ability of the tag-carrying form of YB-1 to be cleaved with the 20S proteasome. The FLAG at the C-terminus may easily change both the YB-1 cleavage ability and cleavage specificity.
The authors are trying (as it seems to me) to extend the conclusions drawn from their results to the data reported by other researchers who used different cell types and different conditions, thereby calling the latter into question, which looks at least premature.
In experiments on the mechanism of YB-1 transition to the nucleus during the cell cycle, mutant two-tag YB-1 with S102, S165, and S176 replaced by alanine was used (Figs. 3 and 4). Such mutants cannot undergo phosphorylation at these sites. It was with these mutants that the authors demonstrated the nuclear translocation of YB-1 at the G2/M boundary instead of G1/S. This experiment had the following drawbacks. First, the residues in question were not replaced by acidic ones, e.g., Asp, imitating the phosphorylated state of YB-1; naturally, the replacement effect on the subcellular distribution of YB-1 was not checked either. Second, all three substitutions were made at a time, not one by one. It is questionable whether phosphorylation/ dephosphorylation of these amino acids occurs simultaneously; also, it cannot be ruled out that the effect of phosphorylation of one of them on YB-1 distribution can be completely different, if not opposite, from that of the others.
Only the cold shock domain of YB-1 is structured; therefore, it is rather difficult to model the structure of the full-length protein. The server I-Tisser offers more than one model of the structure of full-length YB-1. In the manuscript, it is not mentioned on what basis the used model was chosen and what refinement was made. In 2016, an Indian research group offered a model of the structure of full-length YB-1 (Birendra Singh Yadav et al., (2016) Structure prediction and docking-based molecular insights of human YB-1 and nucleic acid interaction, Journal of Biomolecular Structure and Dynamics, 34:12, 2561 2580, DOI: 10.1080 / 07391102.2015.1124050); in this paper, 5 models proposed by I-Tisser were analyzed and refined. Although the authors of the study under review used I-Tisser too, it is very difficult to compare the proposed model with the one published previously, using only figures and having no further description in the manuscript. Meanwhile, justification of the choice and refinement of the model is a fundamental point, since on the basis of these results the authors draw important conclusions that contradict previously reported results. Taking into account the controversial findings and insufficient evidence for the proposed theory/model, the use of another server for protein structure modeling (e.g., Swiss model) could be recommended; this would allow a higher probability of the proposed YB-1 structure.
To Figure 5.
The authors argue that phosphorylation at Ser102, Ser165, Ser174 will alter the accessibility of NLS and CRS. Under Introduction, the CRS motif is described as aa 267-293, while Figure Legend gives aa 247-267. A question arises as to what exactly motif was used in the model analysis. This is of importance because these motifs do not overlap, and it is not clear the accessibility of which motif alters upon phosphorylation.
To Figure 5C, D.
The authors claim that the substitutions S102A, S165A, and S176A lead to YB-1 nuclear localization. It is difficult to draw a conclusion from only 4 cells shown (Figure 5C), but it looks like a significant portion of the protein is still cytoplasmic. The authors should clarify how they classified cells with "nuclear" and "cytoplasmic" localization of YB-1. Besides, the authors should use software for image analysis (measuring the fluorescence intensity in the nucleus and cytoplasm of the cell). To simulate the phosphorylated state, phosphomimetic substitutions (aspartic or glutamic acid) can be used. To support their conclusions, the authors are recommended to make this experiment again using S102D, S165D, and S176D.
The manuscript reads: “Together, these data suggest that the dephosphorylation of YB-1 increases NLS exposure and leads to nuclear localisation, whereas YB-1 phosphorylation increases CRS exposure resulting in increased cytoplasmic presence”. – Actually, this is an invalid conclusion.
Figure S1а
The proteasome cleavage site is shown incorrectly (it is located after NLS, aa 219-220). The CRS motif is shown incorrectly - in fact, it is aa 267-293 (as under Introduction).
Figure S2b
The mass of marker proteins is not indicated. Without it, it is difficult to estimate the size of protein fragments.
Figure S3
The absence of controls - staining for key protein participants of the cell cycle (e.g., cyclins or cyclin-dependent kinases) is necessary to confirm the synchronization and progression through the phases of the cell cycle indicated in the figure.
The manuscript reads “YB-1 remains cytoplasmic as the cells move from G1 through to G2, when dephosphorylation occurs as the nuclear envelope breaks down in late G2, concomitantly reducing the accessibility of the CRS and increasing the accessible energy surface of the NLS resulting in nuclear translocation”. But an active transport through the nuclear membrane can hardly be discussed in the absence of the functional nuclear membrane. Figure 6 needs to be explained or edited: it looks as if upon mitosis, the outer cell membrane disappears and the mitotic spindle forms in the nucleus, with nuclear localization of centrosomes. Besides, the change in phosphorylation during the cell cycle was analyzed for only one residue, Ser102; it would be incorrect to extend this conclusion to Ser165 and Ser174.
Round 2
Reviewer 1 Report
In the revised version the authors have addressed all my concerns and modified the manuscript accordingly.
Author Response
We thank Reviewer 1 for their positive comments regarding the revisions that we have made to the original manuscript. We are pleased that these modifications have now addressed all of their concerns.
Reviewer 2 Report
The revised manuscript has been significantly improved and even though I was unable to download the new version of the supplementary file, I am satisfied by the authors response to the comments and I feel that this study now meets the criteria for publication.
Author Response
We thank Reviewer 2 for their positive comments regarding the revisions that we have made to the original manuscript. We are very pleased that this reviewer believes that our study now meets the criteria for publication. We are unclear as to why the supplementary material was not visible to Reviewer 2; however this material will now be resubmitted with the revised version of the manuscript.
Reviewer 3 Report
Unfortunately, I have not received the revised version of Supplementary Material (instead, there is the original version containing three figures). Therefore, I cannot evaluate some details of the study.
The authors have addressed most of my comments, although some of them remain unanswered. My major concern is the accessibility of the tagged YB-1 for the 20S proteasome. My suggestion was to add a direct control, i.e., to test the ability of the tag-carrying YB-1 to be cleaved with this proteasome in comparison with the wild type YB-1. However, the authors just reversed the tags, which is not the required control.
The absence of this control makes some authors' conclusions questionable.
Thus, I feel that this manuscript still requires major revision.
Author Response
We thank Reviewer 3 for the positive comments that they have made regarding the revisions that we have made in response to the concerns that were raised to the original manuscript. We are thankful that Reviewer 3 has indicated that we have addressed most of their concerns.
1. The outstanding concern that this reviewer has raised is regarding the accessibility of the tagged YB-1 for the 20S proteasome. My suggestion was to add a direct control, i.e., to test the ability of the tag-carrying YB-1 to be cleaved with this proteasome in comparison with the wild type YB-1.
This concern is a genuine one and we thank the reviewer for their diligence in bringing it to our attention. Data showing that that the HA-tag does not interfere with proteasomal cleavage of YB-1 has however already been published (Sorokin et al, EMBO J, 2005). In this paper the authors show that cleavage of ectopically expressed YB-1 by the 20S proteasome produces a 32kDa (and 22kDa) protein species (Figures 2 and Figure 3A) and in Figure 4A this same 32kDa species is generated after incubation of HA-YB-1 with DOX, which is then blocked following treatment with the proteasome inhibitor MG132 (Figure 4D). This HA-tag is the same as that used in our manuscript. These experiments provide clear evidence that the HA tag has no influence on proteasome cleavage.
In further support of this, van Roeyen et al (Cell Communication and Signaling, 2013) report proteasomal cleavage of YB-1 in response to DOX treatment using a fusion protein, where YB-1 was labelled with EGFP. Again, the presence of a tag, in this case much larger than either the HA or FLAG tags used in our manuscript, had no influence on YB-1 cleavage. This also illustrates that neither N-terminal or C-terminal tags interfere with either cleavage or nuclear import.
We have added the following to the manuscript (page 15, lines 6-12) in order to address the concerns of Reviewer 3:
It is also possible that the tags we used (HA and FLAG) could alter putative site-specific cleavage patterns. However, this seems unlikely as Sorokin et al showed that cleavage of ectopically expressed HA-YB-1 by the 20S proteasome produced both 32kDa and 22kDa protein species [7] in untreated and DOX treated cells and the generation of these two species was blocked following treatment with the proteasome inhibitor MG132. Similarly, van Roeyen et al demonstrate that proteasomal cleavage of GFP-tagged YB-1 takes place in response to DOX treatment [9]. Both of these reports illustrate that neither the N-terminal nor C-terminal tags interfere with either cleavage or nuclear import of YB-1.
2. Reviewer 3 was not able to access the revised version of the supplementary material.
We are unclear as to why Reviewer 3 was not able to access the revised supplementary material. We will resubmit it with the revised version of the manuscript. We believe that this supplementary material will sugnificantly improve the manuscript
Round 3
Reviewer 3 Report
The manuscript has been much improved and now warrants publication.